Diversity of lanternfish (Myctophidae) larvae along the Ninety East Ridge, Indian Ocean

Wu Qiong 1 2
Xiang Peng 1
Wang Chunguang 1
Jing Chunsheng 1
Lin Xinyu 1
Wang Yanguo 1
Chen Guangcheng 1 3
Lin Mao 1
Xing BingPeng xingbpeng@gmail.com 1 3 4
1 Third Institute of Oceanography Ministry of Natural Resources , Xiamen , Fujian Province , China
2 College of Life Sciences, Beijing Normal University , Beijing , China
3 Observation and Research Station of Coastal Wetland Ecosystem in Beibu Gulf, Ministry of Natural Resources , Beihai , Fujian Province , China
4 Schmid College of Science and Technology, Chapman University , CA , United States of America
Badenhorst Shaw
Electronic publication date: 2025 Mar 17
Publication date: 2025
Volume: 13
Electronic Location ID: e19144
Received 2024 Oct 21; Accepted 2025 Feb 19
Copyright: ©2025 Wu et al.
Copyright year: 2025
Copyright holder: Wu et al.
License: This is an open access article distributed under the terms of the Creative Commons Attribution License, which permits unrestricted use, distribution, reproduction and adaptation in any medium and for any purpose provided that it is properly attributed. For attribution, the original author(s), title, publication source (PeerJ) and either DOI or URL of the article must be cited.
License URL: https://creativecommons.org/licenses/by/4.0/

Keywords: COI, DNA barcoding, Ichthyoplankton, Larvae, Myctophidae

Funding: National key research and development program 2022YFC3102401 Scientific Research Foundation of Third Institute of Oceanography MNR 2019015 2017009 Global climate change and ocean atmosphere interaction research II GASI-01-NPAC-STsum The Marine Biological Sample Museum GASI-02-YPK-SW This work was supported by the National Key Research and Development Program (grant numbers 2022YFC3102401), Scientific Research Foundation of Third Institute of Oceanography MNR (grant numbers 2019015, 2017009), Global climate change and ocean atmosphere interaction research II (grant numbers GASI-01-NPAC-STsum), and the Marine Biological Sample Museum (number GASI-02-YPK-SW). The funders had no role in study design, data collection and analysis, decision to publish, or preparation of the manuscript.

==============================
Since the 19th century, the impact of seamounts on the distribution of plankton has been a topic of considerable interest. The influence of seamounts on the biogeographic patterns of marine organisms is complex, with some aspects still under debate. It is generally accepted that seamounts can drive the upwelling of nutrient-rich deep waters. Tidal amplification, flow acceleration, and internal waves can further enhance vertical mixing, leading to increased primary productivity near seamounts. Seamounts may also act as barriers to the migration of marine organisms, affecting gene flow. Research on Pacific seamounts suggests these features might serve as “stepping stones” for the dispersal of marine species across the ocean. However, investigations of seamounts in the eastern Indian Ocean remain limited. Focusing on the Ninety East Ridge region in the eastern Indian Ocean, this study collected zooplankton samples using horizontal (surface) and vertical (0–200 m) plankton nets and measured temperature and salinity profiles with a conductivity, temperature, and depth (CTD) sensor. A total of 544 fish larvae were identified, including 260 lanternfish larvae, representing 38 species across 12 genera, determined through COI DNA barcoding. Phylogenetic trees and haplotype networks were constructed to analyze genetic distances and population structures of lanternfish species. Among the samples, intra-specific genetic distances ranged from 0% to 2.99%, while inter-specific distances ranged from 1.88% to 25.71%. Except for Notolychnus valdiviae (Brauer, 1904), the maximum intra-specific distances were lower than the minimum inter-specific distances for all species. Haplotype analysis of nine species revealed significant variations in haplotype number, structure, and spatial distribution. Specifically, Ceratoscopelus warmingii (Lütken, 1892) and N. valdiviae exhibited a notable north-south divergence pattern, consistent with the temperature and salinity distribution of the region’s water masses. This conclusion was supported by analysis of molecular variance analysis, suggesting that larval stages of certain lanternfish species may struggle to cross boundaries between water masses. However, the remaining species showed no significant north-south distribution differences, possibly due to their adaptive capabilities, vertical migration patterns, or the duration of their planktonic larval stages. These findings suggest that seamounts and water mass distribution have varying implications for lanternfish species, potentially influencing gene flow and horizontal distribution patterns, which could contribute to speciation. Global climate change-induced alterations in ocean currents may profoundly impact the genetic diversity of fish species. This study provides new insights into the diversity of lanternfish in the Ninety East Ridge region and offers valuable data for understanding the biogeography of seamounts.

Introduction

Seamounts are widely regarded as critical biodiversity hotspots and play an essential role in supporting biological dispersal by serving as ecological stepping stones (Hanel et al., 2010; Leal & Bouchet, 1991; Wilson Jr & Kaufmann, 1987). They have a significant impact on species composition, productivity, and vertical transfer of nutrients in the surrounding waters (Hubbs, 1959; Lavelle & Mohn, 2010; Morato, Allain & Nicol, 2010; Rowden et al., 2010). Previous studies have demonstrated that seamounts tend to increase the abundance and diversity of fish (Hubbs, 1959; Krishna et al., 2001; Morato, Allain & Nicol, 2010), plankton (Clark et al., 2010; Dai et al., 2020; Dower & Mackas, 1996; Genin, 2004; Mullineau & Mills, 1997), and benthic organisms (Du Preez, Curtis & Clarke, 2016; Pitcher et al., 2008; Richer De Forges, Koslow & Poore, 2000) in their vicinity.

Seamounts can impact ecosystems through a combination of factors (Moore et al., 2003; Rowden et al., 2010), such as upwelling (Hubbs, 1959; Pitcher et al., 2008), vertical spatial heterogeneity (Carney, 2005; Du Preez, Curtis & Clarke, 2016; McClain et al., 2009; White & Mohn, 2004), aggregation of planktonic and planktonic larvae due to blocking currents (Boehlert & Genin, 1987; Krishna et al., 2001; Mullineau & Mills, 1997; Rogers, 1994), substrate heterogeneity (Du Preez, Curtis & Clarke, 2016; McClain, 2007), and the influence of organic matter deposition (Pitcher et al., 2008). Furthermore, seamounts have unique environmental factors, and the distances between them can restrict gene flow (McClain et al., 2009) and result in ecosystems exhibiting distribution patterns similar to those of island organisms (Boehlert & Genin, 1987; Hubbs, 1959; Mullineau & Mills, 1997; Rowden et al., 2010). However, there are significant differences in fauna communities among seamounts in different regions (McClain et al., 2009; Pitcher et al., 2008), and the small coverage of current human investigations on seamounts has led to varying hypotheses about their role in marine ecosystems (McClain, 2007; Rowden et al., 2010; Thoma et al., 2009).

The Ninety East Ridge is located in the northeastern Indian Ocean and is one of the longest volcanic ridges in the world (Krishna et al., 2001; Krishna et al., 2012). The general north-south trend of the ridge extends from 34°S to 17°N (Krishna et al., 2012), with the part of the ridge north of 9°N to 10°N buried in the Bengal fan sediments (Krishna et al., 2012; Sclater & Fisher, 1974; Subrahmanyam et al., 2008). The average width of the ridge is 200 km, and the average height is two km (Bowin, 1973; Krishna et al., 2001; Verzhbitsky, 2003). While the east and west sides of the ridge have similar topographic gradients, the seabed depth is deeper on the east side than on the west side between 11°S and 21°S (Krishna et al., 2001).

Lanternfish are small pelagic and benthopelagic fish, comprising approximately 253 described species (Eschmeyer, Fricke & Van der Laan, 2024), and are among the most abundant vertebrates in the ocean (Poulsen et al., 2013). They contribute over 50–60% of the biomass of deep-sea fish (Farrell, 2011; Martin et al., 2018; Poulsen et al., 2013). The head and body of the fish possess well-developed luminescent organs (Farrell, 2011; Nelson, Grande & Wilson, 2016; Priede, 2017). In a few genera (such as Diaphus), headlight organs play a crucial role in species identification. However, the number and distribution patterns of lateral photophores are more broadly important for distinguishing species across lanternfishes (Helfman et al., 2009; Martin et al., 2018; Moser & Ahlstrom, 1972; Priede, 2017). Lanternfish are known for their diurnal vertical migration behavior, descending to water depths of 300–1,200 m during the day and ascending to depths of 10–100 m at night (Farrell, 2011; Nelson, Grande & Wilson, 2016). The family primarily feeds on plankton, while also serving as prey to various marine organisms. Myctophids is a key component of pelagic food webs (Farrell, 2011; Meincke, 1971) and playing a significant role in the ocean’s carbon cycle (Hudson et al., 2014; Priede, 2017).

Most marine bony fishes have a planktonic larval stage (Cochran, Bokuniewicz & Yager, 2019), and the identification of larval fish is critical for research on species life history, establishing marine protected areas, ecological monitoring, environmental assessment, and formulating fisheries management policies (Maccall, 1979; Moura et al., 2008; Senina et al., 2016; Valdez-Moreno et al., 2010). In ichthyoplankton samples, lanternfish larvae are usually one of the most common components (Batta-Lona et al., 2019; Muhling, Lamkin & Richards, 2012). However, as fish at different developmental stages have different morphological characteristics (Ko et al., 2013), the appearance of individuals at early developmental stages of many taxa is very similar (Valdez-Moreno et al., 2010; Victor et al., 2009), thus it is difficult to accurately identify fish larvae by morphological characteristics. As a result, many taxa can only be identified to the family level using morphological characters (Ko et al., 2013; Wan & Zhang, 2016; Xing et al., 2022). According to Shao, Yang & Chen (2001), only about one-tenth of the known fish can be identified by morphological characters. Although the melanophore pattern of lanternfish larvae can provide a morphological characteristic basis for identification, identifying them still relies heavily on the experience of the identifier, and the large genus Diaphus is particularly difficult to morphologically identify (Richards, 2005). In addition, the high morphological variability of lanternfish larvae further complicates their identification (Sabatés & Saiz, 2000).

The application of genetic technology has emerged as a powerful tool for the identification of fish larvae. Hebert et al. (2003) and Hebert, Ratnasingham & De Waard (2003) first proposed the concept of DNA barcoding and the use of Cytochrome Oxidase I (COI) sequences for species identification. The conserved nature of the COI sequence makes it possible to distinguish samples at the species level for most taxa. The absence of introns, high copy number, and matrilineal inheritance make it one of the most widely used barcoded gene fragments (Bingpeng et al., 2018; Dasmahapatra & Mallet, 2006; Savolainen et al., 2005; Triverdi et al., 2016). The rapid growth of COI sequence data in the databases since 2003 has provided the necessary resources for accurate identification of fish larvae (Porter & Hajibabaei, 2018). According to Ward, Hanner & Hebert (2009) and Ward et al. (2005), COI DNA barcoding can identify most fish species, and Ko et al. (2013) has pointed out that the DNA barcoding technique is one of the best methods for identifying fish larvae. However, there are a few genera, such as Thunnus, where DNA barcoding may be less effective due to factors like high gene flow or significant interspecific introgression (Vinas & Tudela, 2009). Furthermore, the lack of reference sequences is another factor limiting the broader application of DNA barcoding (Chu et al., 2019). Despite these limitations in certain cases, DNA barcoding has now been shown to distinguish lanternfish samples at the species level and even at the geographic population level (Pappalardo et al., 2015; Poulsen et al., 2013).

The diversity of larval fishes in the Ninety East Ridge has received little attention in previous studies (Zhang et al., 2021). To better comprehend seamount ecosystems in the eastern Indian Ocean and gain further insights into the role of seamounts in biological evolution, it is crucial to investigate the diversity of larval fishes in this region. Specifically, this study aims to: (1) assess the diversity and distribution patterns of lanternfish larvae across different regions of the Ninety East Ridge; (2) determine whether DNA barcoding can provide reliable species identification at the larval stage, overcoming challenges posed by morphological similarities; and (3) explore how the findings contribute to understanding the ecological role of seamounts in shaping marine biodiversity. By addressing these questions, we aim to provide new insights into the contribution of seamounts to biological dispersal and the evolutionary processes of marine organisms in the eastern Indian Ocean.

Materials & Methods

Study area and sample collection

Larval samples used in this study were obtained from zooplankton collections conducted in the summer of 2021 in the Ninety East Ridge region of the Indian Ocean, as depicted in Fig. 1. The sampling area, situated to the north of the equator and adjacent to Sumatra Island in the northeast, covered approximately 20 latitudes (9.980°S–24.996°S, ∼1170 nautical miles) from north to south and 5 longitudes (86.601°E–91.417°E, ∼285 nautical miles) from east to west.

Figure 1 Map of sample collection sites.

Background plot from the General Bathymetric Chart of the Oceans (GEBCO) (available at https://www.gebco.net/).

In this study, “larval fish” refers to individuals from the preflexion stage to the juvenile stage. Sampling was conducted using two vertical towing methods and one horizontal towing method. The two vertical towing methods followed the Chinese National Standard (GB/T 12763.6-2007), with the specific procedures described as follows: (1) WP2 plankton net, towed vertically to collect samples over 200 m, with a mouth diameter of 57 cm and mesh size of 0.198–0.202 mm, equipped with a mechanical flow meter (Hydro-Bios); (2) Large plankton net, towed vertically to collect samples over 200 m, with a mouth diameter of 80 cm and mesh size of 0.505–0.507 mm, equipped with a mechanical flow meter (Hydro-Bios); (3) Horizontal trawling using large plankton net, with a trawling time of 0.5–1 h and speed of 0-1 knots. However, due to the slow speed of the flowmeter, it was unable to accurately count, hence, the samples captured in this way were only used for qualitative analysis. As per the sampling plan limitations, not every station in the figure was sampled. A total of 14 stations were sampled using the WP2 plankton net, 17 stations were sampled using the large plankton net, and 19 stations were sampled using horizontal trawling with a large plankton net. Details of the sampling methods and stations are presented in Table 1.

Table 1 Sampling stations characteristics and sampling method.

No.	Station	Sampling date	Sampling method	
1	SEI-03	21.01.18	H	
2	SEI-04	21.01.19	H	
3	SEI-05	21.01.23	L/H	
4	SEI-06	21.01.25	H	
5	SEI-08	21.01.24	L/H/W	
6	SEI-10	21.01.28	L/H/W	
7	SEI-11	21.02.05	L/H/W	
8	SEI-12	21.02.05	L/H	
9	SEI-14	21.02.04	L/H/W	
10	SEI-16	21.02.03	L/W	
11	SEI-19	21.02.02	H	
12	SEI-21	21.01.24	H/W	
13	SEI-22	21.02.08	L/H/W	
14	SEI-24	21.01.23	L/W	
15	SEI-25	21.01.23	L/H/W	
16	SEI-29	21.02.06	L/H	
17	SEI-30	21.01.29	H/W	
18	SEI-32	21.01.29	L/H/W	
19	SEI-33	21.02.01	H	
20	SEI-35	21.01.31	H	
21	SEI-37	21.01.30	H	
22	SEI-38	21.01.30	L/W	
23	SEI-A1	21.02.08	L	
24	SEI-A2	21.02.07	L/H/W	
25	SEI-A3	21.02.07	L/W	
26	SEI-A4	21.02.07	L/W	
Notes.

W WP2 net, towed vertically over 200 m

L Large plankton net, towed vertically over 200 m

H Large plankton net, towed Horizontally

Samples collected using the above-mentioned methods were placed in a collection bottle, and transferred to a sample tube. They were fixed by adding 95% alcohol or RNAhold® (Transgen Biotech) and stored in a refrigerator at 4 °C. Photographs of the samples were taken using a Leica S9D microscope.

DNA extraction and amplification

For DNA extraction, the DNeasy Blood & Tissue Kit (QIAGEN, Shanghai, China) was used following the manufacturer’s protocol for animal tissues. Amplification of the COI mitochondrial sequence was performed using two pairs of primers: 5′-LCO1490: GGTCAACAAATCATAAAGATATTGG-3′, HCO2198: 5′-TAAACTTCAGGGTGACCAAAAAATCA-3′ (Vrijenhoek, 1994); dgLCO1490: 5′-GGTCAACAAATCATAAAGAYATYGG-3′, dgHCO2198: 5′-TAAACTTCAGGGTGACCAAARAAYCA-3′ (Leray et al., 2013). A 25 µL PCR reaction system was prepared containing 12.5 µL PCR mixture (Taq polymerase plus Master Mix II (Dye Plus)), one µL of each primer (10 µM), 2.5 µL DNA template, and eight µL of pure water. The PCR reaction conditions were annealing at 45 °C, increasing by 0.5 °C per cycle for 15 cycles, 49 °C for 20 cycles, resulting in a target product of approximately 690 bp. Samples that met the sequencing concentration were subjected to bidirectional sequencing.

Some samples failed to amplify via PCR, and those that were successfully amplified were sent to Sangon Biotech Co., Ltd (Shanghai, China) for PCR cleaning and sequencing.

Molecular data processing and analysis

Molecular data processing and analysis were performed as follows. Sequences were joined, aligned, and trimmed using SeqMan v. 7.1.0 (DNAStar, U.S.A.) (Burland, 1999), removing low-signal ends to produce a final length of ∼650 bp. Phylogenetic analyses were performed using PhyloSuite (Zhang et al., 2020). Multiple sequences were aligned using the ‘auto’ strategy in MAFFT (Katoh & Standley, 2013). One sequence obtained from GenBank was considered as an outgroup (Scopelengys tristis). The best-fit evolutionary models were determined by selecting for Bayesian Information Criterion (BIC) using ModelFinder (Kalyaanamoorthy, Haeseler & Jermiin, 2017). Maximum likelihood (ML) phylogenies were inferred using IQ-TREE (Nguyen et al., 2015) for 2000 standard bootstraps, as well as the Shimodaira–Hasegawa-like approximate likelihood-ratio test (Guindon et al., 2010). Phylogenetic trees were visualized and edited in iTOL (available at https://itol.embl.de/), following Letunic & Bork (2021).

Pairwise distance analyses were conducted by Kimura-2-parameter (K2P) model (Kimura, 1980) in MEGA11 (Tamura, Stecher & Kumar, 2021), and all ambiguous positions were removed. In order to minimize experimental error resulting from different sampling ranges, only samples captured during this cruise were selected for genetic distance analysis.

Samples were identified using the Basic Local Alignment Search Tool (BLAST) against the GenBank database to confirm their species. Samples with a sequence similarity greater than or equal to 98% (percentage identity) to a known species were considered to belong to the same species as the reference sequence in GenBank (Leray et al., 2013; Machida et al., 2009). To ensure the reliability of the results, reference sequences were selected based on the following principles: preference was given to sequences from published articles, and sequences of the same species uploaded by different authors (except for species with only one sequence in the GenBank database) were used with priority. For samples with a sequence similarity below 98%, the best-matched sequences in the GenBank were downloaded for further analysis, and their taxonomic status was confirmed based on their phylogenetic relationships with known species sequences. Larval fish samples that did not belong to the family Myctophidae were excluded from this study based on molecular identification methods.

Haplotype network graphs were constructed for all species with sample sizes larger than 10. The sequences from each species were aligned using the MAFFT (Katoh & Standley, 2013) plugin available in the Phylosuite software (Zhang et al., 2020). Haplotype diversity statistics were calculated using the DnaSP software (Librado & Rozas, 2009). A haplotype TCS network graph was created using PopART (Population Analysis with Reticulate Trees) (Leigh & Bryant, 2015). Since PopART can only accommodate up to 10 colors (representing collection locations), to avoid color repetition, some color adjustments were made using Adobe Illustrator.

Population structure was assessed using the Arlequin 3.5.2.2 software (Excoffier & Lischer, 2010) through the analysis of molecular variance (AMOVA) method. The fixation index (Fst) (Slatkin, 1995) was calculated, and the demographic history was investigated using mismatch distribution, Tajima’s D (Tajima, 1989), and Fu’s Fs (Fu, 1997) values, all analyzed within the same software.

Population history was inferred using Extended Bayesian Skyline Plots (EBSP) analysis, performed with BEAST v2.7.5 (Bouckaert et al., 2019). The COI gene mutation rate was set to the average mutation rate for ray-finned fishes, 0.015 per million years (May et al., 2020). The nucleic acid substitution model was selected as the HKY model, chosen for its relatively few parameters (Nanini et al., 2024). The Markov chain length was set to 5,000,000. The results of the EBSP were visualized using R v4.3.1 (R Core Team, 2023), following the method outlined by Heled (2010).

Results

Morphology of the specimen

The photographs of the specimens are presented in Figs. S1–S36. The specimens of Diaphus lucidus (Goode & Bean, 1896) and Symbolophorus evermanni (Gilbert, 1905) are not included due to their poor state of preservation and the resulting low reference value of the photographs. It is worth noting that to extract DNA, our samples were preserved using alcohol and RNAhold instead of formalin. This preservation method has significant adverse effects on the morphological identification characteristics of the samples, particularly body length (Chu et al., 2019). Therefore, the photographs in the supplementary figures are provided for reference only.

Species abundance and sample composition

In this study, a total of 544 larval fishes were collected and COI sequences were successfully amplified for 519 of them. Of these, 260 were myctophids, comprising 50% of the total number of larval fish samples. The sequences were submitted to GenBank (http://www.ncbi.nlm.nih.gov), with details provided in Table S1. Of all myctophid samples, 53 samples were collected at 14 stations using wp2 plankton net (vertical) with individual density of 0.0666 ind/m3 on average. Seventy-seven samples were collected at 17 stations using large plankton net (vertical) with individual density of 0.0381 ind/m3 on average. The rest of the samples were collected by horizontal trawl.

In this study, the examined myctophid larval samples belong to a total of 12 genera and 38 species. Among these species, 16 of them remain undefined or unidentified. Ceratoscopelus warmingii (Lütken, 1892) had the highest number of samples among species with 31, while Diaphus had the highest number of samples among genera with 109. Out of the 26 sampling stations, samples of myctophids were not collected at four stations. The species composition of each sampling station is presented in Figs. 2–3.

Figure 2 The species composition and abundance of lanternfish larvae at each station. Black dots denote abundance (right axis).

(A) wp2 plankton net, vertical. (B) Large plankton net, vertical.

Figure 3 The species composition of lanternfish larvae at each station (large plankton net, horizontal).

Phylogenetic trees and genetic distances

According to the BIC, the best-fit model was GTR+F+R5. The resulting phylogenetic tree is presented in Fig. 4.

Figure 4 Phylogenetic tree based on mitochondrial sequences (COI).

The analysis was carried out using Bayesian Inference under the GTR+F+R5 substitution model. The black numbers near nodes are bootstrap support values; only those higher than 85% are indicated. The samples obtained from this study all start with “F” and are not labeled with the species name, while the reference sequences obtained from GenBank are labeled with the species name and GenBank accession number. Branches marked with a pentagram indicate sequences with <98% similarity to known species sequences.

In this study, the intraspecific genetic distances among sample sequences ranged from 0−2.99%, with an average of 0.32%. Meanwhile, interspecific genetic distances ranged from 1.88%–25.71%, with an average of 18.28%. Except for Notolychnus valdiviae (Brauer, 1904), the maximum intraspecific genetic distances were smaller than the minimum interspecific genetic distances for all species. The genetic distance increases significantly as the taxonomic rank increases.

Population structure

The TCS haplotype network graph for nine species with sample sizes greater than 10 was constructed based on COI sequences. The analysis revealed significant variations in the number and distribution patterns of haplotypes across different species (Fig. 5). Notably, C. warmingii and N. valdiviae exhibited significant geographic distribution patterns: the two species were divided into northern and southern populations, with SEI-11 station serving as the boundary (Fig. 6), while the remaining species did not show any distinct geographic distribution patterns. Meanwhile, C. warmingii and N. valdiviae exhibited the highest nucleic acid diversity (Pi > 0.005) among all samples. The haplotype network revealed limited gene flow between the geographically distinct populations of C. warmingii and N. valdiviae (Figs. 5G, 5I). No haplotypes were found to be shared between the populations of these two species. Furthermore, the genetic structure of C. warmingii appears to be more intricate compared to that of N. valdiviae. Based on the haplotypes, the samples can be divided into three populations: the southern population (S), and populations 1 and 2, which together form the northern population (N) (Fig. 5I). Population 1 and population 2 combined to form population N. Population S and population N exhibited distinct distribution ranges, with no shared haplotypes observed between different sampling stations. On the other hand, population 1 and population 2 experienced a minimum of three mutations, but their distribution ranges showed significant overlap, suggesting a potential population expansion subsequent to a brief period of isolation. Among all the samples, the haplotype diversity (Hd) of the nine species ranged from 0.3778 to 0.9869. Diaphus mollis Tåning, 1928 had the lowest Hd, while Diaphus richardsoni Tåning, 1932 had the highest.

Figure 5 Haplotype network from COI sequences, obtained from the TCS analysis.

The diameter of the circles represent the frequency of each haplotype. Mutational steps are symbolized by short line, and black dots mark missing steps. (A) Diaphus brachycephalus (B) Diaphus fragilis (C) Lampanyctus nobilis (D) Diaphus perspicillatus (E) Diaphus richardsoni (F) Diaphus mollis (G) Notolychnus valdiviae (H) Lampanyctus sp4. (I) Ceratoscopelus warmingii.

Figure 6 Distribution and phylogenetic relationships of northern (N) and southern (S) populations of Ceratoscopelus warmingii and Notolychnus valdiviae.

Red dots represent northern populations, while blue dots represent southern populations. The phylogenetic trees illustrate the genetic clustering of northern and southern groups. (A) Ceratoscopelus warmingii (B) Notolychnus valdiviae.

The AMOVA analysis revealed a significant genetic differentiation between the northern and southern water masses in the case of C. warmingii and N. valdiviae. Specifically, C. warmingii showed an Fst value of 0.7245 (P < 0.01), while N. valdiviae exhibited an Fst value of 0.9693 (P < 0.01). These findings highlight the substantial genetic differentiation observed between the northern and southern populations of both species.

Historical demography

Given the limited sample sizes for most species, this study focused on the demographic history of two species, C. warmingii and N. valdiviae, where geographic population differentiation was more pronounced.

The neutrality test analysis data are presented in Table 2. The results revealed that in C. warmingii, both the northern and southern populations exhibited Tajima’s D and Fu’s Fs values below 0, although the significance of Tajima’s D was not observed. Regarding N. valdiviae, while the northern population exhibited a Fu’s Fs value greater than 0, only the southern population displayed a significant Fu’s Fs value (P < 0.05).

Table 2 Neutrality tests for North and South populations of Ceratoscopelus warmingii and Notolychnus valdiviae in the Ninety East Ridge.

Species	Populations	Tajima’s D test	Fu’s FS test	
		Tajima’s D	Tajima’s D p-value	FS	FS p-value	
Ceratoscopelus warmingii	N	−1.04775	0.138	−13.2145	0	
S	−0.50439	0.332	−3.45327	0.006	
Mean	−0.77607	0.235	−8.33389	0.003	
Notolychnus valdiviae	N	−0.65405	0.326	0.1098	0.462	
S	−1.23311	0.112	−1.81298	0.019	
Mean	−0.94358	0.219	−0.85159	0.24	

The mismatch distribution of both the north and south populations of C. warmingii and N. valdiviae showed inconsistencies with expectations (Fig. 7). N. valdiviae for all populations mismatch distribution displayed a distinctive bimodal pattern with distant peaks, splitting the northern and southern populations to show single peaks. The northern population of C. warmingii showed multiple peaks, and the southern population had single peaks, suggesting that both species may be experiencing population expansion. Initially, the multiple peaks in the mismatch distribution of the northern population of C. warmingii were thought to be due to its two distinct subpopulations (Pérez-Portela, Villamor & Almada, 2010), but even after sample separation, each subpopulation’s mismatch distribution continued to exhibit multiple peaks.

Figure 7 Mismatch distribution.

(A) Ceratoscopelus warmingii (all populations) (B) C. warmingii (Northern population) (C) C. warmingii (Southern population) (D) Notolychnus valdiviae (all populations) (E) N. valdiviae (Northern population) (F) N. valdiviae (Southern population).

EBSP analysis results indicated population expansions for C. warmingii and N. valdiviae approximately 0.1 million years and 0.04 million years ago, respectively (Fig. 8).

Figure 8 Extended Bayesian skyline plots showing population size changes over time.

(A) C. warmingii; (B) N. valdiviae.

Furthermore, attempts were made to perform EBSP analyses on different geographic populations of C. warmingii and N. valdiviae using BEAST v2.7.5. However, despite adjusting the priori parameters and increasing the chain length, the Effective Sample Size (ESS) for these populations remained below 200, precluding reliable conclusions from being drawn.

Discussion

Morphology of the specimen

As one of the most ubiquitous fish groups in the marine environment, lanternfish larvae are among the most taxonomically tractable and extensively studied of all fish larvae (Moser, 1996). Most myctophid larvae exhibit distinct morphological and pigment characteristics that facilitate morphological identification (Moser, 1996; Richards, 2005). The head, body, and gut shapes of the larvae typically show similarity within genera (Moser, Ahlstrom & Paxton, 1984).

In our lanternfish samples, 16 species could not be identified to the species level due to a lack of reference data in the GenBank database. For these species, we primarily relied on comparisons of their morphology with descriptions provided by Moser, Ahlstrom & Paxton (1984), Moser (1996) and Richards (2005). After excluding species with existing COI data in the database, the morphology of Lampanyctus sp.2 closely resembled Lampanyctus acanthurus Wisner, 1974 (Fig. S27). However, Moser (1996) noted that Lampanyctus acanthurus typically has a pigment spot anterior to the adipose fin, which was absent in our specimen, and there were differences in the shape of the head pigment spots. Additionally, no distribution records of Lampanyctus acanthurus in the Indian Ocean were found. Therefore, we tentatively designate our sample as an unidentified species.

Another specimen with morphology similar to reference descriptions is Lampanyctus sp.1 (Fig. S26). Although poorly preserved, the head morphology closely matches the description of Lampanyctus pusillus (Johnson, 1890) by Moser, Ahlstrom & Paxton (1984). DNA barcoding also confirmed that this specimen belongs to the genus Lampanyctus. However, as Lampanyctus pusillus already has reference data in GenBank, and our COI sequence shares less than 90% similarity with it, this specimen was designated as an unidentified species.

Although lanternfish larvae are among the most extensively studied fish larvae (Richards, 2005), the number of species that can be identified morphologically remains limited compared to adult fish, particularly in species-rich genus Diaphus. Morphological identification is notably challenging for this genus (Moser, 1996). For example, in the Subtropical Convergence Zone, at least 80 myctophid species are known, but larvae have been described for only 60 of them (Bolshakova & Prokofiev, 2024).

This situation highlights a paradox: while DNA barcoding is a powerful tool for linking larval and adult stages, morphological identification retains advantages in speed and cost. However, the preservation methods commonly used often fail to balance the maintenance of well-preserved morphological features with the ability to extract DNA.

Species abundance and sample composition

Zooplankton abundance in the Indian Ocean region is closely associated with monsoons and displays substantial seasonal variation (Koné et al., 2009; Veldhuis et al., 1997). Following the onset of the southeast monsoon, algal blooms appear and zooplankton abundance peaks in the following one to two months. Previous studies have indicated that decapods, amphipods, ichthyoplankton, and cnidarians are most abundant in the eastern Indian Ocean region during winter (June–September) and lowest in early summer, with intermittent sub-peaks in early autumn (March). Horizontally, the abundance of plankton in this area tends to decrease with increasing latitude between 9°S and 32°S, while the ichthyoplankton abundance reaches its peak between 24°S and 25°S (Tranter & Kerr, 1977). The productivity of the Ninety East Ridge region surpasses that of surrounding areas, mainly because of the impact of monsoons and seamounts (Brewer et al., 2015). However, due to variations in the shape and size of seamounts, as well as the horizontal spatial scale of seamounts being comparable to that of biological fields in the ocean, it is challenging to draw precise conclusions from a single marine biological survey (Pitcher et al., 2008).

Lanternfish are among the most common fish in marine environments (Richards, 2005), and collecting data on their larvae is a prerequisite for understanding the ecosystem of Ninety East Ridge region. We utilized the COI DNA barcoding technique to identify the larvae of lanternfish in the Ninety East Ridge waters during the summer season. A total of 38 species of larvae from 12 genera were recognized, representing 15% of the known living lanternfish species. Among them, the genus Diaphus had the highest number of samples and is also the largest genus within the family Myctophidae (Elloran, 2012; Eschmeyer, Fricke & Van der Laan, 2024). Stations SEI-24, SEI-25, and SEI-A2 had the highest abundance of larval fish, with lower abundances observed at southern stations. The distribution pattern of lanternfish larvae resembles that of the zooplankton described by Tranter & Kerr (1977), but differs from that of the ichthyoplankton mentioned in the same study.

The density of individuals captured by the large plankton net was lower than the WP2 net, probably due to the larger mesh size, which missed some smaller-sized larvae. Conversely, the total number of individuals and species captured by the large net was higher than the WP2 net, likely due to the smaller mouth diameter of the WP2 net, which resulted in inadequate sampling. Additionally, since not all stations utilized all three sampling methods and the horizontal tows lacked quantitative data, differences in sampling methods may have introduced bias into the results.

Out of all the sampling stations, lanternfish samples were not collected at four stations. These four stations were exclusively sampled using horizontal trawls. This may be related to the shallow water depth (∼1 m) of horizontal trawling sampling. According to previous literature, lanternfish larvae exhibit a lower abundance in the surface waters (Loeb, 1979; Tsukamato et al., 2001). This could be due to the significant variability in temperature and salinity in surface waters (Tsukamato et al., 2001), or their behavior to avoid solar radiation, which has been shown to have harmful effects on larvae (Alves & Agusti, 2020; Miller & Kendall Jr, 2009).

The samples included 16 unidentified species. Despite our attempts at morphological identification, we were unable to confirm the species to which these samples belong. Determining the taxonomic status of these samples will require further studies on adult myctophids in the Ninety East Ridge region. The diversity of lanternfish in this area may still warrant further exploration.

Phylogenetic trees and genetic distances

The phylogenetic tree demonstrates that COI DNA barcodes can effectively differentiate most species at the species level. Notably, for species such as N. valdiviae, C. warmingii, Diaphus parri Tåning, 1932, Diaphus splendidus (Brauer, 1904) and Lampanyctus nobilis Tåning, 1928, COI barcodes can distinguish between different samples at the population level. However, the monophyly of the genus Benthosema appears problematic as Benthosema glaciale (Reinhardt, 1837) and genus Diogenichthys form the same crown group. This observation aligns with the findings of Martin et al. (2018), Denton & Adams (2015), and De Busserolles et al. (2014). Martin et al. (2018) proposed that Benthosema and Diogenichthys form a subclade, with Benthosema being paraphyletic. Our phylogenetic tree exhibits a topology similar to that of Martin et al. (2018), and we suggest that it would be more appropriate to place Benthosema glaciale within the genus Diogenichthys. However, since the taxonomic status of Benthosema glaciale has not yet been formally revised, Fig. 1 continues to use Benthosema as the genus name for this species.

The COI sequences obtained in this study differed somewhat from the sequences of D. effulgens obtained from GenBank. However, the similarity was greater than 98%, leading to the conclusion that they belong to the same species (Machida et al., 2009). Further study may be necessary to confirm the validity of this identification, particularly after capturing adult fish with sequences more similar to the samples.

Although most samples in this study had maximum intraspecific genetic distances smaller than the minimum interspecific genetic distances, there were significant differences in interspecific genetic distances among different genera. For instance, the average interspecific genetic distance of genus Benthosema was 16.94%, genus Diaphus was 11.20%, and genus Lampadena was only 5.03% (Table S2). It is possible that there may not be a universal threshold for interspecific genetic variation that can be applied across all genera in the family Myctophidae to successfully distinguish species.

Population structure

Hd is a measure of the uniqueness of a haplotype within a specific population, reflecting haplotype abundance in that population. Pi, on the other hand, measures the degree of polymorphism within a population (Quan et al., 2020). The haplotype diversity (Hd) and nucleotide diversity (Pi) of the nine analyzed lanternfish species showed substantial variation (Fig. 5). C. warmingii and N. valdiviae were the only two species with Pi > 0.005, and both also exhibited high Hd values (>0.75). This indicates that these two species possess rich genetic diversity within the sampled populations, which could be attributed to a large effective population size (Hague & Routman, 2016), a high mutation rate (Amos & Harwood, 1998), or habitat heterogeneity (Amos & Harwood, 1998), such as adaptations to different temperature-salinity water masses or secondary contact between two populations (Grant & Bowen, 1998).

In contrast, Diaphus fragilis Tåning, 1928, and D. mollis had Pi values lower than 0.001. The Hd of D. fragilis was 0.5909, and that of D. mollis was 0.3778. High Hd combined with low Pi may suggest that these two species experienced a historical population bottleneck, with the current populations resulting from a rapid expansion of small founder populations.

C. warmingii has long been recognized for its significant intraspecific genetic differentiation (Bernard et al., 2022; Gordeeva, 2011; Santos, 2021). This study further strengthens these previous findings by revealing a remarkable similarity in the geographic population distribution patterns of both C. warmingii and N. valdiviae within the Ninety East Ridge. Notably, both species demonstrate a distinct differentiation into two populations based on the northern and southern water masses. This differentiation closely aligns with the thermohaline distribution patterns of these two water masses at depths ranging from 100 to 200 m (Fig. 9). Additionally, Isaji et al. (2022) reported that, apart from differences in temperature and salinity, significant variations in dissolved oxygen and nitrate concentrations are also observed at a depth of 200 m in this region.

Figure 9 Distribution of temperature and salinity levels in investigated areas.

In this study, C. warmingii exhibited a complex haplotype network. Among the 26 identified haplotypes, only two were found to be distributed across multiple populations. Interestingly, both of these haplotypes were classified as interior haplotypes, referring to haplotypes located at the internal nodes of haplotype networks, which are shared by multiple individuals or lineages. This suggests that these haplotypes may have existed prior to the divergence of the two populations.

Fst is a measure of population differentiation due to genetic structure. An Fst value > 0.15 is considered indicative of significant differentiation between populations (Luo et al., 2019). In this study, the Fst values for C. warmingii and N. valdiviae were both substantially higher than 0.15, with N. valdiviae showing an exceptionally high Fst value of 0.9693. This indicates that the genetic variation within these species is primarily driven by genetic differences between the northern and southern populations, with minimal gene flow between them. In contrast, genetic variation within each population is low, suggesting frequent gene exchange within populations. Both species are globally distributed and relatively common, but their intraspecific genetic diversity and its underlying causes warrant attention. Dos Santos et al. (2024) identified seven highly structured clusters within C. warmingii, with divergence times traced back to the Miocene through the Pleistocene. Our findings suggest that the study area may represent a boundary zone between two of these clusters. Although data on the distribution of adult lanternfish in this region are lacking, making it unclear whether adults can traverse these water masses, the differences in larval distribution hint at potential ecological niche differentiation. This could be linked to the separation of spawning grounds or differing adaptations to the temperature and salinity conditions of distinct water masses. Similar phenomena, where water masses act as barriers to genetic exchange between populations, have been documented in various taxa (Goldson, Hughes & Gliddon, 2001; Johannesson et al., 2018; Planes, Parroni & Chauvet, 1998). Our findings provide evidence that this phenomenon also occurs in lanternfish. The extinction of local populations with distinct genotypes can erode the genetic diversity of these species and reduce their potential to adapt to climate change (Manel et al., 2020). Therefore, for fish species with complex genetic structures, more nuanced conservation strategies are needed.

The larval stage of fish is often highly susceptible to environmental factors, and it remains unclear whether the observed divergence between the two species is due to distinct adaptations to thermohaline factors or limited movement, resulting in gene dispersal solely within the respective water mass. The distribution of fish larvae is typically influenced by a combination of biological and environmental factors (Moser & Smith, 1993). The spawning location of adult fish, the availability of food resources, and the dispersal effects of water currents all play significant roles in shaping the distribution patterns of larvae (Mullaney & Suthers, 2013). Water masses, in particular, have a crucial influence on the distribution of larval fish (Olivar & Beckley, 2022; Tiedemann et al., 2018). Given that climate change can impact ocean circulation (Keller et al., 2000; Phillips et al., 2021; Rahmstorf, 2002), thermohaline circulation (Clark et al., 2002) and distribution of water masses (Stanev, Peneva & Chtirkova, 2019), it is essential to closely monitor the effects of global climate change on the genetic diversity of fish species. Due to the sensitivity of larvae to environmental changes, the widespread distribution of lanternfish, and their critical role in the marine food web (Meincke, 1971), the dynamics of lanternfish larvae may hold significant potential for monitoring changes in marine environments. Additionally, as seamount ecosystems are known to aggregate economically important fish species on their flanks (Pitcher et al., 2008), they are subject to pressures from commercial fishing (Schlacher et al., 2010). Therefore, when establishing marine protected areas and drafting fisheries policies, the conservation of genetic diversity in ichthyoplankton populations should be carefully considered.

In other species with number of samples greater than 10, such as D. brachycephalus Tåning, 1928, D. fragilis, D. perspicillatus (Ogilby, 1898), D. richardsoni, and Lampanyctus sp.4, no discernible north-south population divergence distribution pattern was observed. For instance, the haplotype network map indicates that D. brachycephalus is centered around Hap_3, with the rest of the haplotypes distributed radially, indicating that the species may have undergone a rapid population expansion, and the distribution range of Hap_3 haplotypes include multiple stations from south to north. It is speculated that the differences in the geographical distribution patterns of haplotypes in different species may be linked to variations in the length of the larval planktonic period and different vertical migration patterns. Although almost all adult myctophids exhibit diurnal vertical migration, some species have different vertical migration patterns for larvae and adult fish (Richards, 2005), and various subfamilies have different vertical distribution patterns for larvae (Sassa et al., 2004). These variations in vertical migration behaviors among species and subfamilies can potentially contribute to differences in population structure for these organisms.

SEI-29 appears to have a high specificity for several species, as it contains the most haplotypes for C. warmingii, D. brachycephalus, D. richardsoni, and Lampanyctus nobilis. Additionally, for C. warmingii, D. brachycephalus, D. fragilis, D. mollis, D. perspicillatus, D. richardsoni, Lampanyctus nobilis, and N. valdiviae, interior haplotype distribution was observed at SEI-29. Meincke’s (1971) study on the Great Meteor Seamount demonstrated that the anticyclonic vortex of isolated seamounts can trap plankton. SEI-29 is also situated above a relatively isolated seamount. It is possible that some haplotype-specific larvae drifted in from the southwest or further out and were subsequently captured and concentrated at this location. Another explanation for the high haplotype diversity at SEI-29 is that it is simply the result of dispersal. In addition, local ecological conditions, such as substrate diversity and differences in hydrological conditions, may also influence the distribution of larvae (Hanafi-Portier et al., 2024). To determine which hypothesis is correct, data from the area surrounding the station, particularly from the southwest, are required.

Historical demography

Reconstructing population history dynamics typically necessitate the integration of neutrality tests and mismatch analyses. In the present study, the outcomes of neutrality tests conducted on C. warmingii were incongruous: Fu’s Fs indicated a population expansion, while Tajima’s D did not yield significant results. This discrepancy may be attributed to a relatively recent population expansion or an insufficient sample size (Simonsen, Churchill & Aquadro, 1995). According to Simonsen, Churchill & Aquadro (1995), an ideal analysis would include more than 50 samples. Previous research by Ramos-Onsins & Rozas (2002) demonstrated that Fu’s Fs tend to be more effective than Tajima’s D in detecting population changes. Moreover, the results of the EBSP analyses provide supporting evidence for population expansion in C. warmingii (Fig. 8). Therefore, there is a growing inclination to believe that this species has indeed undergone population expansion.

The neutrality test results for N. valdiviae present a contradictory picture, as Fu’s Fs indicates that only the southern population has undergone expansion. However, the mismatch distribution analysis reveals a distinct bimodal pattern with distant peaks across all populations. This pattern may be related to the low gene flow between the two populations (Ray, Currat & Excoffier, 2003). Moreover, when dividing the populations into northern and southern groups, a single peak is observed, suggesting that both populations have experienced population expansion (Rogers & Harpending, 1992). The EBSP analyses also support the occurrence of population expansion in N. valdiviae, although the extent of expansion appears to be more moderate compared to C. warmingii. However, the low ESS observed in the EBSP analyses for each geographic population of both species may result from a limited number of mutation sites within populations or insufficient sample sizes.

Conclusions

In summary, this study utilized COI barcodes to identify 38 species of lanternfish larvae in the Ninety East Ridge, Indian Ocean. The effects of different net types on sampling results were compared. The construction of a phylogenetic tree and the analysis of haplotype networks revealed that different lanternfish larvae exhibit distinct distribution patterns. For most myctophid species, the Ninety East Ridge and water masses do not appear to impede gene flow. However, for C. warmingii and N. valdiviae, water masses may act as barriers to genetic exchange between geographically distinct populations. This conclusion is further supported by the results of AMOVA analyses. These findings suggest that the distribution of water masses may promote the divergence of distinct evolutionary lineages in these two species, either by restricting gene flow or by exerting selective pressures on larvae through differences in temperature and salinity. Additionally, various analyses including neutrality tests, mismatch analyses, and EBSP analyses suggested that both species may have undergone population expansions, although the timing of these expansions may not be consistent. Among the species investigated in this study, the SEI-29 station consistently displayed the highest aggregation of haplotypes. This observation could potentially be attributed to the station’s location on a relatively isolated seamount in the area. The topography of the seamount likely plays a significant role in shaping gene exchange and determining the population distribution of larval fish.

Supplemental Information

Supplemental Information 1 260 NCBI GenBank accession numbers obtained in this study

Supplemental Information 2 Pairwise intraspecific and interspecific genetic distances of lanternfish samples

Supplemental Information 3 Photograph of lanternfish larval specimen

We wish to thank Nathalie Yonow for her invaluable advice. We thank Delara Rustam for her advice and help with data processing. Special thanks are due to all of the group members that helped us during the sample collecting. We utilized ChatGPT-4o to review grammar, tense, and spelling errors in our submission. Following its use, we have thoroughly verified the accuracy of all content.

Additional Information and Declarations

Competing Interests

Author Contributions

Field Study Permissions

DNA Deposition

Data Availability

The authors declare there are no competing interests.

Qiong Wu performed the experiments, analyzed the data, prepared figures and/or tables, authored or reviewed drafts of the article, and approved the final draft.

Peng Xiang conceived and designed the experiments, authored or reviewed drafts of the article, and approved the final draft.

Chunguang Wang conceived and designed the experiments, performed the experiments, prepared figures and/or tables, authored or reviewed drafts of the article, and approved the final draft.

Chunsheng Jing performed the experiments, analyzed the data, prepared figures and/or tables, and approved the final draft.

Xinyu Lin analyzed the data, prepared figures and/or tables, and approved the final draft.

Yanguo Wang performed the experiments, prepared figures and/or tables, sample Collection, Sample Processing, and Financial Support, and approved the final draft.

Guangcheng Chen conceived and designed the experiments, prepared figures and/or tables, sample Collection, Sample Processing, and Financial Support, and approved the final draft.

Mao Lin conceived and designed the experiments, authored or reviewed drafts of the article, and approved the final draft.

BingPeng Xing performed the experiments, authored or reviewed drafts of the article, and approved the final draft.

The following information was supplied relating to field study approvals (i.e., approving body and any reference numbers):

The samples for this experiment were collected from scientific research-based trawling in the open ocean, with specimens selected from plankton samples. They do not fall within the scope requiring approval from an Ethics Committee, nor are they subject to the regulations of the Chinese Laboratory Animal Welfare Law (GB/T35892-2018).

The following information was supplied regarding the deposition of DNA sequences:

The 260 accessions are available in GenBank (Table S1).

The following information was supplied regarding data availability:

The data that support the findings of this study are openly available in GenBank, reference numbers are in Table S1.

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
