# Peer review of "Diversity of lanternfish (Myctophidae) larvae along the Ninety East Ridge, Indian Ocean"

_PeerJ, doi:10.7717/peerj.19144_

## Round 0.1 · original submission · Major Revisions

The reviewers highlighted several issues that the authors must address.

Reviewer 1 ·

Basic reporting

Overall, accept with minor revisions
Abstract
The abstract addresses the impact of seamounts on plankton distribution, specifically focusing on the Ninety East Ridge region in the eastern Indian Ocean. It highlights the collection and analysis of 260 lanternfish larvae representing 38 species and utilizes advanced molecular techniques (COI DNA barcoding) to assess genetic variability and species distribution. The abstract effectively conveys the genetic and distributional insights into lanternfish larvae from the Ninety East Ridge, emphasizing molecular approaches and specific species patterns. However, it could be improved by providing greater ecological context, methodological clarity, and discussion of broader implications. This would make the study more accessible and impactful for a wider scientific audience.
However, here are some weak points:
1. The abstract does not sufficiently explain why seamounts, specifically, are hypothesized to affect plankton and lanternfish distribution. Adding background on seamount ecology or currents in the Ninety East Ridge region would strengthen the argument.

Provide a brief explanation of the ecological significance of seamounts as biodiversity hotspots and their interaction with planktonic communities.

2. While genetic results are detailed, the ecological significance of observed haplotype variations and the divergence pattern is underexplored in the abstract.

Include additional details about the methodology (e.g., sampling depths, barcoding protocols).

3. While COI barcoding is mentioned, other supporting methods (e.g., environmental sampling techniques, statistical analyses) are not discussed.

Elaborate on the ecological or evolutionary significance of the divergence patterns, particularly the north-south split in species like C. warmingii and N. valdiviae.

4. The focus on Ceratoscopelus warmingii and Notolychnus valdiviae seems disproportionate without further elaboration on their ecological importance.

Clarify whether the findings are novel or support existing hypotheses about seamount effects on plankton distribution.

5. Generalized conclusions. The linkage between water masses and gene flow is a significant claim but requires more context to understand the mechanisms driving this relationship.
Discuss broader implications of these findings for conservation efforts or understanding regional marine biodiversity.

Experimental design

The methods section provides a detailed description of the sampling, laboratory procedures, molecular analysis, and data processing steps used in this study on larval lanternfish in the Ninety East Ridge. It effectively balances thoroughness with clarity, making it reproducible and comprehensive for readers. Below is a breakdown of its strengths, weaknesses, and areas for improvement. The methods section is highly detailed, rigorous, and technically sound, reflecting a well-designed study. However, it could benefit from clearer explanations of methodological choices, a discussion of potential biases, and a more concise presentation of technical details. These changes would enhance accessibility while maintaining the section’s scientific robustness.
1. While the methodologies are well-described, the rationale for certain choices is unclear. For instance: Why were specific primers or sampling depths (200m) chosen? AND Why was horizontal trawling used only for qualitative analysis?
Explain the selection of sampling methods (e.g., depth, mesh sizes) and statistical models to show how they align with the study objectives. Justify the choice of molecular markers and specific experimental parameters, such as the COI mutation rate.


2. While models such as HKY, K2P, and Bayesian skyline plots are mentioned, their relevance to the biological questions could be elaborated upon.

3. The choice of the COI gene mutation rate (0.015 per million years) lacks justification, leaving the reader wondering if this value is standard or specific to the taxa studied.
4. The uneven sampling across stations (14 for WP2 net, 17 for large net, and 19 for horizontal trawling) may introduce bias, but this is not addressed.
Address potential biases due to uneven sampling across stations or the inability to quantify horizontal trawling data. Suggest how these limitations may affect the study’s conclusions.

5. While the section is well-cited, some references (e.g., Burland, 1999; Heled, 2010) are not directly explained, which may leave non-specialist readers unclear on their relevance.
Streamline the phylogenetic analysis description, focusing on its relevance to the study objectives rather than technical minutiae.

6. The detailed discussion of evolutionary models and bootstrap methods, while informative, may be excessive for some readers. A more streamlined summary could suffice.
Provide more details on how PopART visualizations were adjusted in Adobe Illustrator to ensure transparency. Briefly outline how the different analyses (e.g., haplotype networks, AMOVA, and Bayesian skyline plots) collectively address the research questions.

Validity of the findings

6. The section on morphology is underdeveloped. While it acknowledges poor preservation of some specimens, there is little interpretation of the morphological data presented in the supplementary figures.
No connection is made between morphology and genetic findings, missing an opportunity to explore phenotypic-genotypic correlations.

Provide more detailed descriptions of the morphological features observed in well-preserved specimens, particularly highlighting any unique traits linked to species differentiation or adaptation to the seamount environment.

7. While these species exhibit significant population dynamics, the results section disproportionately focuses on them, leaving little room for discussion of the other 36 species and 12 genera sampled.
Include a brief discussion of the genetic and population dynamics of other genera and species analyzed. Summarizing key patterns or anomalies among the remaining species would give a more holistic view.

8. The inability to draw conclusions from EBSP analyses due to low ESS for certain populations is a limitation that could have been better addressed by considering alternative approaches or emphasizing other analyses.
Link the observed genetic and haplotype diversity patterns to specific environmental factors (e.g., water mass distribution, currents) in the Ninety East Ridge.

9. The results lack a strong ecological context, particularly in linking observed genetic differentiation and population structures to environmental or oceanographic factors of the Ninety East Ridge.
Add explanations for genetic diversity indices like Hd and Pi, and consider using visual aids (e.g., box plots) to compare these metrics across species or populations.

10. The section mentions haplotype diversity (Hd) and nucleotide diversity (Pi) but does not explain their ecological or genetic significance in enough detail.
Discuss the challenges encountered during EBSP analyses and suggest how future studies could overcome these limitations (e.g., by increasing sample sizes or employing alternative methods).

11. The description of populations (e.g., “population S,” “population N”) could benefit from clarification about their geographic or ecological boundaries.
Summarize key observations from figures (e.g., Fig. 5, Fig. 6) in the text to ensure readers can interpret results without constant reference to supplementary materials.interpret results without constant reference to supplementary materials.

Additional comments

Discussion
12. The discussion briefly mentions differences between zooplankton and ichthyoplankton distribution patterns but does not delve into the potential mechanisms driving these differences. For example, how might seamount dynamics or oceanographic features of the Ninety East Ridge contribute to these patterns?

13. Elaborate on the specific ways in which seamounts might create favorable conditions for lanternfish larvae, such as through nutrient upwelling, habitat complexity, or aggregation of prey species.

14. While the productivity of the Ninety East Ridge is attributed to monsoons and seamounts, the discussion does not elaborate on the specific role of seamounts in influencing lanternfish larval distribution.

15. Discuss potential reasons for the observed differences between zooplankton and ichthyoplankton distribution patterns, considering environmental gradients, predator-prey relationships, or larval transport mechanisms.

16. The presence of 16 unidentified species is acknowledged, but there is no discussion of what challenges (e.g., morphological variability, lack of reference sequences) may hinder their identification.

17. While the section acknowledges differences in capture rates between nets, it could further analyze how these biases might influence species composition or abundance estimates.

18. The ecological and conservation implications of the findings are not fully addressed. For instance, how might the observed diversity and abundance of lanternfish larvae inform ecosystem management or regional biodiversity assessments?

19. Provide insights into why the 16 unidentified species remain unresolved. Highlighting limitations in DNA reference databases or morphological challenges would add depth to the discussion.
20. *Discuss the relevance of these findings to broader ecological and conservation themes, such as their role in pelagic food webs, contributions to the carbon cycle, or the need for biodiversity monitoring in seamount regions.

21. *Suggest follow-up studies to address knowledge gaps, such as taxonomic studies on adult myctophids, investigations into larval ecology, or broader regional surveys.

22. While the discussion attributes genetic divergence to water masses acting as barriers, it does not explore alternative factors, such as behavioral isolation or larval retention mechanisms, in enough depth.

23. Expand on potential factors beyond thermohaline circulation that could influence genetic differentiation, such as larval retention zones or adult migratory behaviors.

24. The reliance on thermohaline distribution patterns as the primary driver could be better supported with additional data or literature references.

25. Provide brief definitions or explanations for technical terms, such as "ancestral polymorphism," to ensure accessibility for readers less familiar with population genetics.

26. Terms such as "ancestral polymorphism" and "internal haplotype distribution" are used without sufficient explanation, which could confuse readers unfamiliar with population genetics.

27. Compare and contrast the dispersal and vortex trapping hypotheses more rigorously, considering additional ecological or genetic data that could support one over the other.

28. The section speculates on potential mechanisms driving haplotype diversity at SEI-29 but does not critically evaluate these hypotheses. For example, the dispersal hypothesis could be contrasted more directly with the vortex trapping theory.

29. While the section briefly mentions the relevance of climate change, it could elaborate on how the observed genetic patterns might inform conservation strategies, such as protecting vulnerable populations or regions affected by changing ocean currents.

30. Elaborate on how projected changes in ocean circulation and thermohaline structure might affect the genetic diversity and population connectivity of lanternfish species in the future.
31. The text describes complex genetic and geographic patterns, but no accompanying figures or maps are directly referenced. For example, visualizing the haplotype networks or geographic distributions of populations could enhance clarity.
Include or reference figures, such as geographic maps, phylogenetic trees, or haplotype networks, to help readers visualize the patterns described in the text.

32. While the reliance on Fu’s Fs over Tajima’s D is justified, the discussion does not explore why Tajima’s D failed to detect significant population changes. Potential explanations, such as sampling limitations or demographic history nuances, are absent. Explain why Tajima’s D might fail to detect population expansion, perhaps due to factors like a recent expansion event or selective pressures. A brief discussion of these possibilities would provide balance.

33. For N. valdiviae, the discussion moves quickly from describing the bimodal mismatch distribution to asserting population expansion without fully addressing why this pattern might be indicative of such an event. Elaborate on how the bimodal pattern and single-peak grouping are reconciled with population expansion hypotheses, addressing potential alternate explanations (e.g., admixture or stable population structure).

34. The section mentions secondary contact as a potential explanation for high genetic diversity but does not explore alternative or complementary mechanisms, such as historical connectivity or migration events between the northern and southern populations.

35. Incorporate specific details from the EBSP analyses, such as estimated timing and intensity of population expansions, to contextualize the inferred demographic histories.

36. The discussion of EBSP results lacks specific details about the timing or magnitude of population expansion. Without numerical or graphical references, the interpretation remains abstract.
Discuss additional factors that might contribute to high haplotype and nucleotide diversity, such as larval dispersal patterns, historical gene flow, or genetic drift in isolated populations.

37. Discuss additional factors that might contribute to high haplotype and nucleotide diversity, such as larval dispersal patterns, historical gene flow, or genetic drift in isolated populations. Reference relevant figures or visualizations (e.g., EBSP plots, mismatch distribution graphs) to help readers connect textual interpretations with data.

Annotated reviews are not available for download in order to protect the identity of reviewers who chose to remain anonymous.

Reviewer 2 ·

Basic reporting

The manuscript is fairly well-written but contains inaccuracies, lack of important citations, and has some issues with data representation.

Minor/Line-by-line suggestions
Line 1: In the title, should it not be the “Indian Ocean” instead of the India Ocean?

Line 20: Authors state 38 species from 11 genera, but in the results they state 12 genera.

Line 63: The authors should just look up the current number of species and cite the proper authority. (Fricke, R., Eschmeyer, W. N. & R. van der Laan (eds) 2024. ESCHMEYER'S CATALOG OF FISHES: GENERA, SPECIES, REFERENCES.(http://researcharchive.calacademy.org/research/ichthyology/catalog/fishcatmain.asp). Electronic version accessed dd mmm 2024.) There are 253 currently described and accepted species

https://researcharchive.calacademy.org/research/ichthyology/catalog/SpeciesByFamily.asp?_gl=1*1nosyux*_gcl_au*NDUyODU0NjMxLjE3MzE5Mzc4NTc.*_ga*NjMzODkzNDc0LjE3MzE5Mzc4NTc.*_ga_6Y72VP61VZ*MTczMTkzNzg1Ni4xLjAuMTczMTkzNzg1Ni42MC4wLjA.

Line 64-65: Authors need different citations for ‘abundance’ and ‘widespread,’ or at least sentence needs to be revised. As written it just states their diversity as the cause of being abundant.

Line 67-68: This statement is flat-out false. The authors need to seriously spend some time in their literature review. The books the cite for this statement do not support the authors claim. Lanternfishes do NOT have bacterial bioluminescence.

Line 68-69: The headlight organs are only crucial for identification of species within specific genera (like Diaphus). The patterns of the numerous lateral photophores are MUCH more important for species identification across lanternfishes.

Line 71: Space between 1200 and m.

Line 72: Space between 100 and m.

Line 75: Space prior to (

Lines 88-92: The authors need to access and read through Moser, Ahlstrom, and Paxton’s studies on lanternfish larvae. Although not specifically targeting species from the Indian Ocean, many lanternfishes occur globally and the work by these researches dives DEEP into larval identification of lanternfishes (the two listed below at a minimum). This manuscript, which focuses on larval identification of lanternfishes, sorely lacks information from these truly important studies of the larvae on this group… We agree that marine larvae can be very hard to identify and this study using COI is justified, but there are decades of work by researchers on that very topic, that need to be included and addressed.

Moser, H. G., Ahlstrom, E. H., & Paxton, J. R. (1984). Myctophidae: development. Ontogeny and systematics of fishes, 1, 218-239. https://swfsc-publications.fisheries.noaa.gov/publications/CR/1984/8462.PDF

Moser, H. G. (Ed.). (1996). The early stages of fishes in the California Current region. US Department of the Interior, Minerals Management Service, Pacific OCS Region.
https://books.google.com/books?hl=en&lr=&id=vGcWAQAAIAAJ&oi=fnd&pg=PP1&ots=6O5rkXf0hV&sig=J_dYL3cBgFDnl5sofSJyDv2dlCM

Line 102: & should be changed to and

Line 107: Lanternfish shouldn’t be capitalized

Line 116: Would reword to state the larvae samples were from and not ‘were based on’

Line 117: India Ocean should be changed to Indian

Lines 123-125: Make sure to add spaces between numbers and m

Line 133: What was in the collection bottle? Water, ethanol, ?

Line 195: Add space between years and (

Line 197: R citation is out of date for this version

Line 207: Would suggest changing ‘experiment’ to ‘study’

Line 209: Reword “The sequences of them were…”

Line 212: Write out 77 if beginning a sentence with a number.

Line 215: The abstract stats 11 genera

Line 220: Fig(s)

Line 241: Capitalize Population after N.

Line 246: Write out 9

Line 252: Please add those P values

Line 270: space between subpopulations and (

Line 291: Lanternfishes are not considered the most common, bristlemouths are considered to be more abundant.

Line 292: Change to ‘their’ larvae

Line 293: Combine lantern and fish

Line 294: Is it 11 genera, or 12? Both have been mentioned.

Line 296: Would cite the most recently updated taxonomy for (Fricke et al., 2024) See above.

Line 307: Space between 1 and m

Line 310: Was there any attempt at identification not using genbank? I mention above the multiple publications (and there are more than the two suggested) that talk about larval lanternfish identification). I would highly suggest the authors make an attempt at identification of these, given they have chosen to include them in the phylogeny. Especially since the phylogeny may give the authors a good indication of what genus to look at for identification. I understand the condition of some specimens may not allow for this, but I would suggest an attempt for any that are in relatively good condition.

Line 315: The authors add taxonomic authors to some of the species here but did not do so for species first mention in the manuscript above. I would either add the taxonomic authors at first mention of species above and later in the manuscript below (e.g., Scopelengys tristis, Ceratoscopelus warmingii, Notolychnus valdiviae) or remove them here for consistency.

Lines 317-318: This has been found in numerous phylogenies of lanternfishes, which should be discussed and cited here.

Line 357: Space after factors and before (

Lines 359-362: Spaces before (

Line 383: Edit this in-text citation

Figures 2-4: What do the blue dots mean? I would suggest the authors change some of the colors, the blues are hard to distinguish between each other.

Figure 5: Myctophum obtusirostre is now considered to be Dasyscopelus obtusirostris sensu Martin et al. (2018) and accepted by (Fricke et al., 2024). The grey lines on the graph make it difficult to see the phylogeny in some cases. I would increase the line width of the phylogeny for easier reader viewing. Lastly, would you mark on the phylogeny (possibly with a star) which individuals met your 98% similarity criteria and were considered a particular species?

Supplementary Images: Please list, when possible, the FXXX identification name next to each larvae image so readers can determine which image belongs to which taxa in the phylogeny. I would additionally suggest the authors add images of the 16 unknown species (a number that may decrease depending on whether the authors are able to identify them based on morphology as I suggested earlier).

Throughout the manuscript, would make sure species in a sentence are in alphabetical order (ex: line 365)

Other publications of note for the authors to look at: https://onlinelibrary.wiley.com/doi/full/10.1111/jfb.15218
https://spo.nmfs.noaa.gov/content/identification-guide-larvae-lanternfishes-teleostei-myctophidae-subtropical-convergence

Experimental design

no comment

Validity of the findings

I believe the work in this manuscript has merit and that the information presented will be useful to the scientific community.

Something the authors might want to consider incorporating into the discussion is how the fish diversity of this seamount compares to variation in diversity seen other seamount studies, both of which can be compared to diversity in the pelagic.

Reviewer 3 ·

Basic reporting

There are some Mistakes in the text mostly spelling mistakes and grammatical mistakes. Please take it serious and go through the same. And about some concepts which made me confusing. Please rectify it as everything specifies in text as comments

Experimental design

Research design is ok as this is an attempt and Not much studies are done. Your initiatives are quite good

Validity of the findings

Validity of findings are relevant and will be more strong if you can add more studies to support your findings

Additional comments

Please do the necessary corrections on the manuscript.

Annotated reviews are not available for download in order to protect the identity of reviewers who chose to remain anonymous.

---

## Round 0.2 · Minor Revisions

The authors addressed the comments of the reviewer. However, the reviewer is requesting a few minor changes, which will strengthen the paper.

Reviewer 3 ·

Basic reporting

No comment

Experimental design

No comments

Validity of the findings

Manuscript Review: "Lanternfish Larvae Distribution and Population Dynamics in the Ninety East Ridge, Indian Ocean"
Summary: The manuscript is an interesting and comprehensive study on the genetic diversity and distribution patterns of lanternfish larvae from the Ninety East Ridge, identified using COI barcoding to 38 species. The study compared the effects of different net types on sampling, explored the genetic differentiation of species across water masses, and looked into population expansion patterns for key species, C. warmingii and N. valdiviae. The findings of the study provide valuable information on the role of water masses in shaping genetic diversity and evolutionary lineages in lanternfish species.
Strengths:
The integration of molecular techniques, such as COI barcoding, with ecological sampling provides a robust and thorough approach to studying larval distribution patterns and population structure. Both genetic and ecological data are combined to strengthen the conclusions of the study.
In-depth analysis on genetic patterns may be made on the construction of a phylogenetic tree, haplotype networks, besides AMOVA analyses, giving an effective interpretation of the structure of lanternfish populations. To this extent, the finding about water masses presenting barriers to certain species' gene flow adds new dimensions to such ecological factors associated with genetic differences in the Indian Ocean.
Focus on Population Dynamics: Analysis of population expansion in C. warmingii and N. valdiviae using neutrality tests, mismatch distribution, and EBSP analyses provides informative information on the evolutionary history of these species. Variability in the timing of expansions across species adds nuance to our understanding of their population dynamics.
These studies may have clear implications with respect to the conservation of lanternfish populations, both in the context of climate change and marine protected areas. The study would also be useful in providing a foundation for future monitoring efforts and devising conservation strategies.
Improvements:
Hypothesis Testing for SEI-29: The finding that the aggregation of haplotypes at the SEI-29 station is due to its seamount location on an isolated seamount is intriguing. However, further sampling from surrounding areas-perhaps especially south-west of the station-would strengthen the argument. In fact, it would be helpful to discuss possible alternative explanations for this observation, such as oceanographic current patterns or local ecological conditions.
Discussion of Potential Ecological Implications: The manuscript might further develop the ecological significance of the observed genetic divergence and population structure of lanternfish larvae. For example, how will such divergence between C. warmingii and N. valdiviae affect the roles of each in the marine food web or vulnerability to changes such as those related to climate change?

Additional comments

Minor Comments:
More clearly stating the study's key research questions and objectives in the introduction may help present those study's main questions and objectives more coherently. Overall, the background is pretty well presented; however, it is hard to immediately see what the ultimate research questions are.

Annotated reviews are not available for download in order to protect the identity of reviewers who chose to remain anonymous.

---

## Round 0.3 · accepted · Accept

The authors addressed the last set of minor revisions requested by the reviewer, and the paper can now be accepted for publication.